# A reconfigurable photosensitive split-floating-gate memory for neuromorphic computing and nonlinear activation

Zhi-Cheng Zhang[1], Yuan Li[1], Jian Yao [2], Zhaolong Chen [3], Fu-Dong Wang[1], Shu-Han Si[4], Yue Ding[1], Hui-Ling Qi[1], Tong-Bu Lu [4], Lixing Kang [2] ✉, Zhi-Bo Liu [1,5] ✉, Jian-Guo Tian [1,5] ✉ & Xu-Dong Chen [1,6] ✉

The rapid growth of artificial intelligence and the Internet of Things calls for compact hardware platforms that integrate sensing, computing, and nonlinear processing within a unified architecture. However, most existing neuromorphic systems implement only partial functionalities and rely on heterogeneous device integration, limiting scalability and efficiency. Here, we show a high-speed, reconfigurable multi-modal split-floating-gate memory that monolithically integrates in-sensor computing, in-memory computing, and multiple nonlinear activation functions within a single device structure. By programming charges in spatially separated floating gates, the device enables non-volatile analog control of photoresponsivity and conductance, as well as electrically reconfigurable rectification to emulate ReLU and Sigmoid activations. We further demonstrate a fully hardware-implemented sensor–processor system based on the multi-modal split-floating-gate memory arrays that performs complete unsupervised and supervised learning tasks. This work establishes a compact, energy-efficient, and reconfigurable hardware foundation for scalable intelligent systems beyond conventional silicon architectures.

The proliferation of edge devices in the era of artificial intelligence (AI) and the Internet of Things (IoT) has led to massive generation of unstructured data directly at the sensory interface[1,2]. Conventional vision systems are typically built on a rigid architectural separation between sensing, memory, and processing modules, requiring frequent and energy-intensive data transfer from peripheral sensors to centralized processors[2–4]. This bottleneck severely limits the real-time performance and energy efficiency of edge-intelligent systems. In contrast, biological vision systems process sensory information hierarchically and locally within neural circuits (Fig. 1a), achieving low latency and high throughput through unified sensing–processing pathways[5–9].

To emulate such efficiency, recent advances in in-sensor computing (ISC)[3,5,10–18] and in-memory computing (IMC)[19–25] have enabled analog-domain matrix operations, significantly reducing reliance on data conversion and long-range transfer[1,12,26,27]. While several platforms (e.g., complementary metal semiconductor oxide (CMOS) circuits[28–36], analog-to-digital converters (ADCs)[37–43], and Mott device[44]) have demonstrated individual nonlinear activation functions (NAFs) in hardware, the integration of multiple, reconfigurable activation types

[1]The Key Laboratory of Weak Light Nonlinear Photonics, Ministry of Education, School of Physics, Nankai University, Tianjin, China. [2]Division of Advanced Materials, Suzhou Institute of Nano-Tech and Nano-Bionics, Chinese Academy of Sciences, Suzhou, China. [3]School of Advanced Materials, Peking University Shenzhen Graduate School, Peking University, Shenzhen, China. [4]MOE International Joint Laboratory of Materials Microstructure, Institute for New Energy Materials and Low Carbon Technologies, School of Material Science and Engineering, Tianjin University of Technology, Tianjin, China. [5]State Key Laboratory of Photovoltaic Materials and Cells, Nankai University, Tianjin, China. [6]Academy for Advanced Interdisciplinary Studies, Nankai University, Tianjin, China. ✉e-mail: lxkang2013@sinano.ac.cn; liuzb@nankai.edu.cn; jjtian@nankai.edu.cn; chenxd@nankai.edu.cn

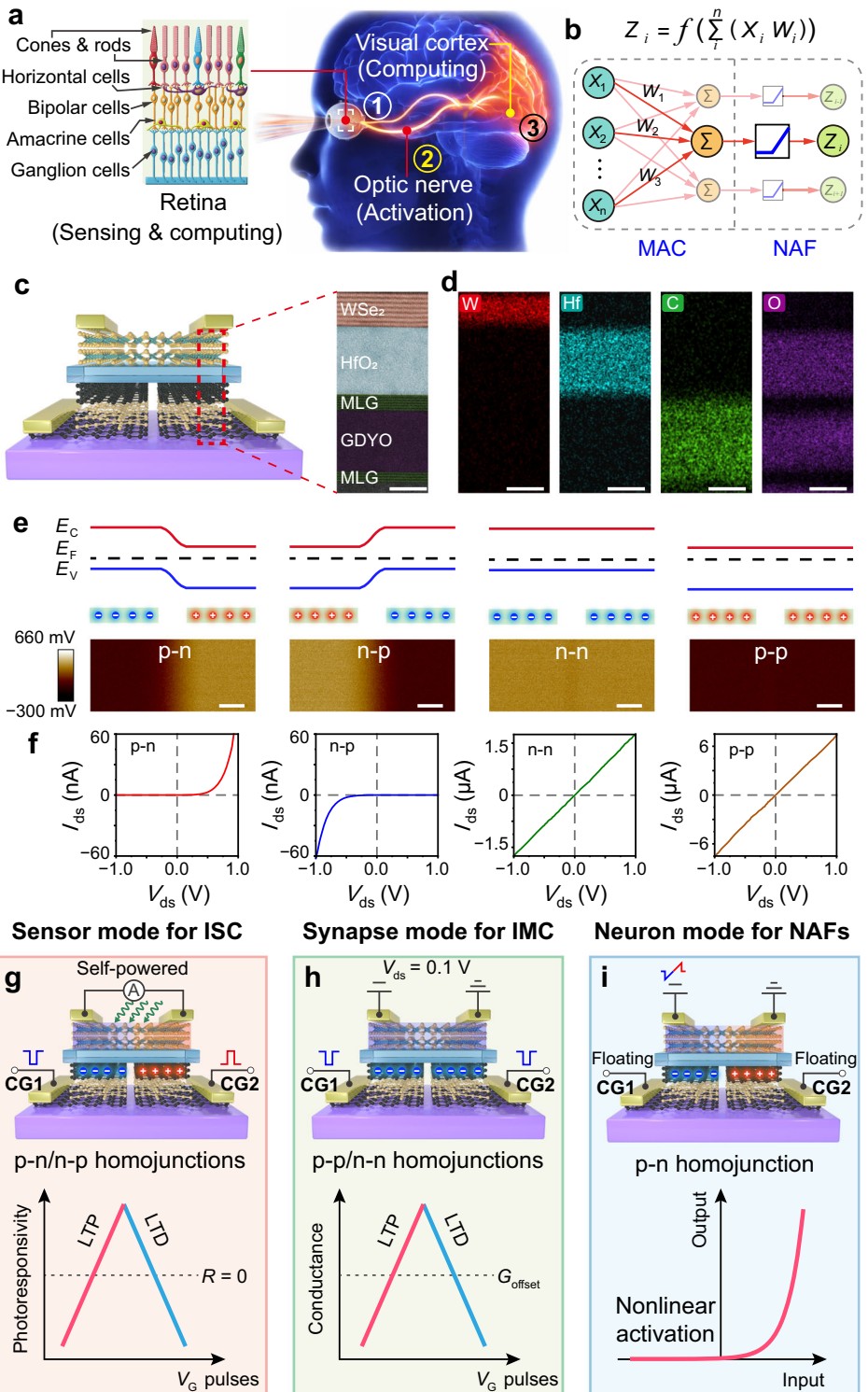

**Fig. 1 | Mechanism of the MM-SFGM with reconfigurable functionalities.**
**a** Illustration of the human visual system consisting of retina, optic nerve, and visual cortex for hierarchical processing. Visual information is first perceived and processed in the retina, where key features are extracted and significantly compressed. These compressed features are then transmitted via the optic nerve to the visual cortex for further, more complex processing in the brain. **b** Illustration of a neural network layer combining linear MAC and nonlinear activation operations in each neuron. **c** Illustration of the structure and cross-sectional scanning transmission electron microscope (STEM) image of the MM-SFGM. Scale bar, 5 nm. **d** Energy dispersive spectrometer (EDS) images for W, Hf, C and O. Scale bars, 5 nm. Kelvin probe force microscopy (KPFM) images (**e**) and corresponding $I$–$V$ curves (**f**) of the homojunction at p-n, n-p, n-n and p-p states, respectively. Scale bars, 1 μm. Configurations of the MM-SFGM operating for ISC (**g**), IMC (**h**) and NAF (**i**), respectively.

(e.g., ReLU and Sigmoid) alongside sensing and computing capabilities within a single physical device remains an unresolved challenge. Moreover, few platforms have succeeded in implementing all core computational primitives—ISC, IMC, and NAFs—on a single material system, a limitation that hinders compactness and prevents full-stack monolithic integration.

Herein, we introduce a high-speed, reconfigurable multi-modal split-floating-gate memory (MM-SFGM) that consolidates ISC, IMC, and tunable NAFs within a unified device structure. By leveraging charge modulation in spatially decoupled floating gates, the MM-SFGM supports linear and non-volatile control over photoresponsivity and conductance, and features electrically programmable junction asymmetry for real-time switching between ReLU- and Sigmoidal-type activation modes. We further demonstrate an end-to-end, fully hardware-based sensor–processor system comprising MM-SFGM arrays that perform both supervised classification and unsupervised autoencoding tasks. Our results demonstrate a scalable platform that eliminates the traditional need to combine multiple device types for different functions, offering a compact, reconfigurable, and general-purpose solution for future sensor–processor hardware beyond neuromorphic or silicon-constrained architectures.

## Results
### Mechanism of the memory with reconfigurable multifunctionality

A reconfigurable MM-SFGM is developed to construct NMVS. The structure of the MM-SFGM, as shown in Fig. 1c, d and Supplementary Fig. 5, consists of an ambipolar $WSe_2$ channel, a $HfO_2$ dielectric layer, and two split multilayer graphene/graphdiyne oxide/multilayer graphene (MLG/GDYO/MLG) heterostructures. As described in Supplementary Note 1 and Supplementary Figs. 11 and 16, the GDYO layer exhibits volatile threshold-switching (TS) characteristics, which is critical for achieving nanosecond-scale programming at low voltages and long-term charge storage in the MLG floating gates[45]. Under a nanosecond-scale negative or positive $V_{CG}$ pulse, the GDYO TS layer transiently switches to a conductive state, allowing direct injection of electrons or holes into the floating gates; once the pulse ends, the TS layer spontaneously returns to the insulating state, confining the charges for long-term non-volatile modulation. In contrast, conventional flash devices rely on tunneling through thick insulating barriers, which also provide non-volatility but typically require higher voltages and longer programming times, leading to slower switching and higher energy consumption. Therefore, the adoption of GDYO is essential to enable the low-energy, high-speed, and multifunctional reconfigurability of the MM-SFGM. Details for the device fabrication is described in Methods and Supplementary Figs. 2 and 3. By controlling the type and number of charges in the split floating gates, different 2D homojunctions (including p-n/n-p/n-n/p-p) can be configured in the $WSe_2$ channel (Fig. 1e and f). Due to the long-term retention of charges in the floating-gates, these configurations exhibit non-volatile characteristics (Supplementary Fig. 17). Therefore, this MM-SFGM can be configured to multiple states, enabling it to perform different tasks.

Specifically, the MM-SFGM can perform ISC tasks when the homojunction is programmed to the p-n and n-p states (Fig. 1g). ISC enables the direct construction of neural networks within the sensor array, facilitating forward-propagation algorithms and convolutional operations using photoresponsivity as the weights[3,10,12,14] (Supplementary Note 2). For ISC, a linear and non-volatile modulation of photoresponsivity ($R$) is required across both positive and negative regimes[3,12,14]. The MM-SFGM operates under short-circuit condition, where self-powered positive/negative photoresponse is realized in the n-p/p-n homojunction via the photovoltaic effect (Supplementary Fig. 18). The photoresponse can be dynamically tuned from positive to negative by applying paired voltage pulses ($-V_{CG}/+V_{CG}$) to CG1 and CG2, and vice versa (Supplementary Fig. 19).

Similarly, when the MM-SFGM is configured to the p-p or n-n states, it can perform IMC tasks using the conductance ($G$) states as the weights (Fig. 1h and Supplementary Note 2)[21,22,46–49]. In this case, the two control gates are connected, and identical $V_{CG}$ pulses are applied to CG1 and CG2. The linear and non-volatile modulation of conductance is achieved by adjusting the number of restricted charges in the split floating gates, which in turn modulates the doping levels of the channel (Supplementary Fig. 26).

Moreover, the rectifying behavior of the MM-SFGM in the p-n state enables the realization of NAFs (Fig. 1i and Supplementary Note 3). By adjusting the electron and hole concentrations stored in the split floating gates, the device exhibits tunable rectification characteristics. When the electron/hole density is low, the device exhibits a ReLU-type $I–V$ curve, while higher densities facilitate the formation of a Sigmoidal-type NAF. Consequently, the neuron-mode MM-SFGM can effectively perform NAFs on the weighted sum results in ANNs.

### Sensor mode of the memory for in-sensor computing

The MM-SFGM can function as a self-powered photodiode when the homojunction is programmed to the p-n or n-p states. Figure 2a shows the spatial distribution of photocurrents generated at the p-n/n-p junction interface without drain bias. By adjusting the stored charges in the split floating-gates, the short-circuit photocurrent ($I_{sc}$) and photoresponsivity can be dynamically modulated between $\pm 80\,mA\,W^{-1}$ (Fig. 2b and Supplementary Fig. 19). Over 63 distinct states (6 bits) are accessible through long-term potentiation (LTP) and long-term depression (LTD) processes, achieved by applying paired $V_{CG}$ pulses with opposite polarities ($\pm 1.2\,V$, 20 ns). The energy required for programming is as low as 4.8 fJ. The nonlinearity, symmetry, and cycle-to-cycle variation (CCV) of the photoresponsivity update are calculated as 0.29 (LTP)/0.23 (LTD), 800, and 3.1% (LTP)/2.9% (LTD), respectively (Supplementary Note 4). These characteristics allow precise control of photoresponsivity during the update process, enabling accurate weight adjustments through the application of the corresponding number of paired $V_{CG}$ pulses to the split control gates, as demonstrated in Fig. 2c.

The photoresponsivity modulation is non-volatile due to the long-term charge storage in the split floating-gates. As shown in Fig. 2d, $I_{sc}$ remains stable across 17 distinct photoresponsivity states for over 1000 s. The endurance of the photoresponsivity during updates is demonstrated in Fig. 2e, where the photoresponsivity is programmed to three exemplary states (0 and $\pm 80\,mA\,W^{-1}$) for $10^6$ cycles, showing negligible deterioration throughout. Additionally, all 63 photoresponsivity states were programmed for 1000 receptions, showing a narrow distribution without no overlap (Fig. 2f and Supplementary Fig. 29). The photoresponse, with both positive and negative photoresponsivity, is shown in Supplementary Fig. 30a and b, with rise and fall time of 450 ns and 290 ns, respectively. Figure 2g and Supplementary Fig. 30 show the photocurrents generated by the device when triggered by optical pulses with different widths. A fully response is observed for pulse widths exceeding 500 ns, indicating the nanosecond operation speed of the MM-SFGM in sensor mode.

In addition to precise photoresponsivity update, the linear dependence of photocurrent on incident light intensity ($P$) is crucial for multiply-accumulate (MAC) and matrix-vector multiplication (MVM) operations in ISC hardware[50]. Unlike the nonlinear power dependence of photocurrents induced by charge-trapping or redox processes[51,52], the $I_{sc}$ generated by the photovoltaic effect in our device exhibits excellent linearity in a quite large power region (Fig. 2h and Supplementary Fig. 31). The analog-analog multiplication capability of the device for ISC was further validated by illuminating the device (with 7 photoresponsivities in range of $\pm 60\,mA\,W^{-1}$) with effective light intensities ranging from 60 to 120 nW. The generated $I_{sc}$ matched the ideal results closely (Supplementary Fig. 32), demonstrating the high accuracy of the $P$-$R$ multiplication. Furthermore, 1000 optical pulses

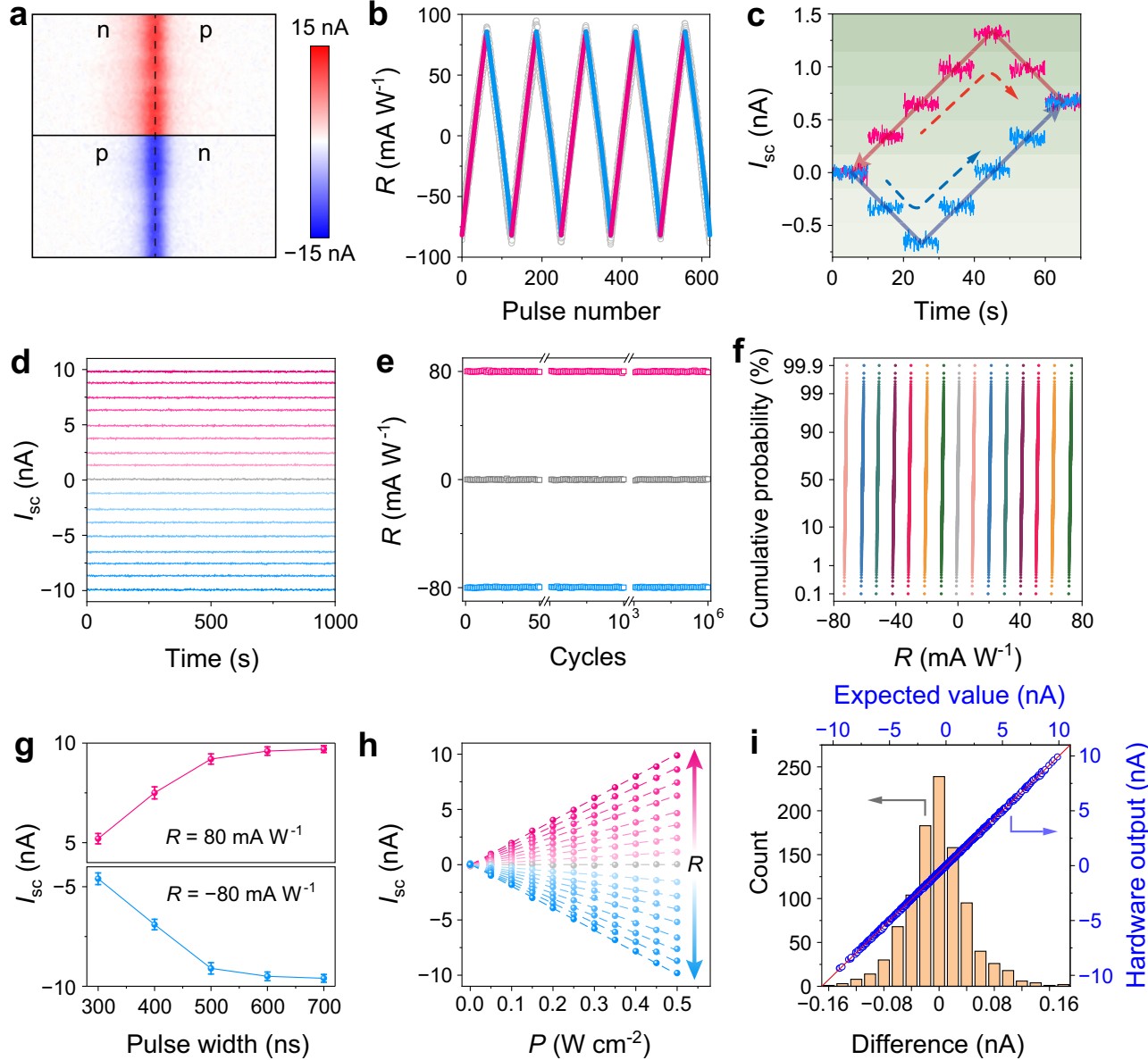

**Fig. 2 | Sensor mode of the MM-SFGM for ISC. a** Photocurrent mapping images of the homojunction at n-p and p-n states. **b** Linear and symmetric photoresponsivity update between ±80 mA W⁻¹ by applying paired $V_G$ pulses with opposite polarities (±1.2 V, 20 ns). **c** Precise modulation of the photoresponsivity between various levels by applying corresponding numbers of paired $V_G$ pulses to the split control-gates. **d** Long-term retention characteristics of the photoresponsivity at 17 distinct states. **e** Cyclic endurance of the photoresponsivity for $10^6$ cyclic test. **f** Cumulative probability distribution of the photoresponsivity with respect to 15 discrete states for 1000 repetitions. **g** Photocurrents of the device with photoresponsivity of 80 mA W⁻¹ (pink) and −80 mA W⁻¹ (blue) while applying an optical pulse with different widths. Error bars denote the standard deviation from 10 measurements. **h** Light-intensity dependence of the photocurrents for the device at 17 distinct photoresponsivity states. **i** Top and right axes: Linear fitting between the hardware output photocurrents and expected values of the analog-analog multiplications for 1000 randomly generated light intensities (0−125 nW) and photoresponsivities (−80 to 80 mA W⁻¹). Bottom and left axes: Statistical distribution of the current differences between experimental and ideal results.

with randomly generated intensities were applied serially to the MM-SFGM, and the hardware outputs ($I_{sc}$) were in good agreement with the expected results (Fig. 2i). Noteworthily, since the dark current of each photodiode under short-circuit condition is ultralow, the generated positive and negative photocurrents can accumulate directly without subtracting dark currents (Supplementary Fig. 33).

## Synapse mode of the memory for in-memory computing

The IMC algorithm, capable of executing MVM operations within a memory array, is considered an ideal platform for neuromorphic computing, requiring linear and symmetric conductance updates across a wide range[22,53,54]. As discussed in Supplementary Note 2, the

MM-SFGM can perform IMC when the entire channel is programmed between the intrinsic undoped state (low conductance) and the heavily p-doped state (high conductance). The doping level and conductance are controlled by the number of charges stored in the floating-gates (Supplementary Fig. 26). As shown in Fig. 3a and b, the device demonstrates a linear, symmetric, and non-volatile conductance update within the range of 0.1−7.0 μS. The robust endurance of conductance update is demonstrated for over $10^6$ cycles (Fig. 3c), and the precise modulation of the conductance to the target levels during the update process can be realized by controlling the number of applied $V_{CG}$ pulses (Supplementary Fig. 34), simplifying the hardware implementation of weight updating. Figure 3d and Supplementary Fig. 35

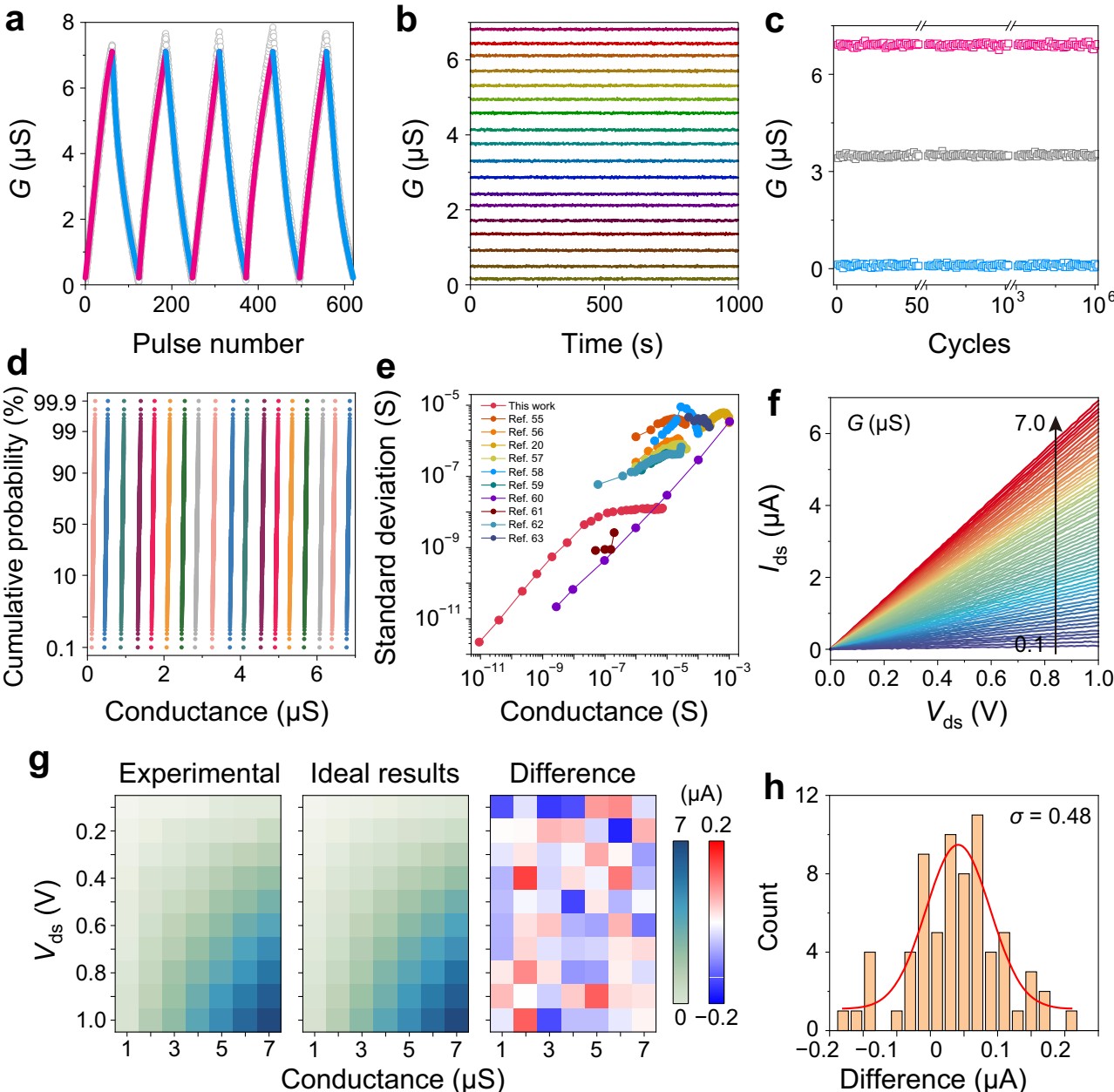

**Fig. 3 | Synapse mode of the MM-SFGM for IMC. a** Linear and symmetric conductance updating in range of 0.1–8.0 μS by applying ±1.2 V/20 ns $V_G$ pulses. **b** Long-term retention characteristics of the conductance at 18 distinct states. **c** Cyclic endurance of the conductance for $10^6$ cyclic tests. **d** Cumulative probability distribution of the conductance with respect to 18 discrete states for 1000 repetitions. **e** Standard deviation (SD) of conductance as a function of conductance, measured over a range from 10 pS to 7 μS. Each conductance state was programmed ten times and read one hundred times after each write, with the SD calculated from the distribution of read values. Representative data from previous reports[20,55–59,103–106] are included for comparison. **f** Linear I–V curves of the device with 63 monotonically potentiated conductance states. **g** Analog-analog multiplications between different pairs of input voltages (vertical axis) and device conductance (horizontal axis). The left and middle panels show the experimental and ideal output currents, respectively, while the right panel represents the current differences between the experimental and ideal results. **h** Statistical distribution of the current differences as presented in (**g**).

show the cumulative probability distribution of the MM-SFGM with 63 discrete conductance states for 1000 receptions, where all the states are well separated without overlapping. Beyond the cumulative distribution analysis within the linear region, we further evaluated the stability of 30 representative conductance states covering the entire dynamic range from 10 pS to 7 μS. Repeated programming (10 cycles) and reading (100 times per write) demonstrated standard deviations consistently below 13 nS (Fig. 3e and Supplementary Fig. 36), confirming the device's excellent stability and reliability across all operation levels. Notably, the standard deviation (SD) increases nearly linearly with conductance in the low-conductance regime

(10 pS–0.1 μS), whereas it saturates in the high-conductance regime (0.1 μS–7 μS), indicating that the device retains stable performance even at large conductance levels. To benchmark our results, Fig. 3e includes comparative data from several representative reports[20,55–59] that exhibit narrower conductance ranges and significantly higher SD values. In contrast, the MM-SFGM maintains an ultra-wide conductance window and exceptionally low fluctuation, demonstrating excellent noise suppression and reliability compared with state-of-the-art memory and neuromorphic devices.

MVM operations inherently require a linear relationship between the output current and the input voltage. Figure 3f shows the linear I–V

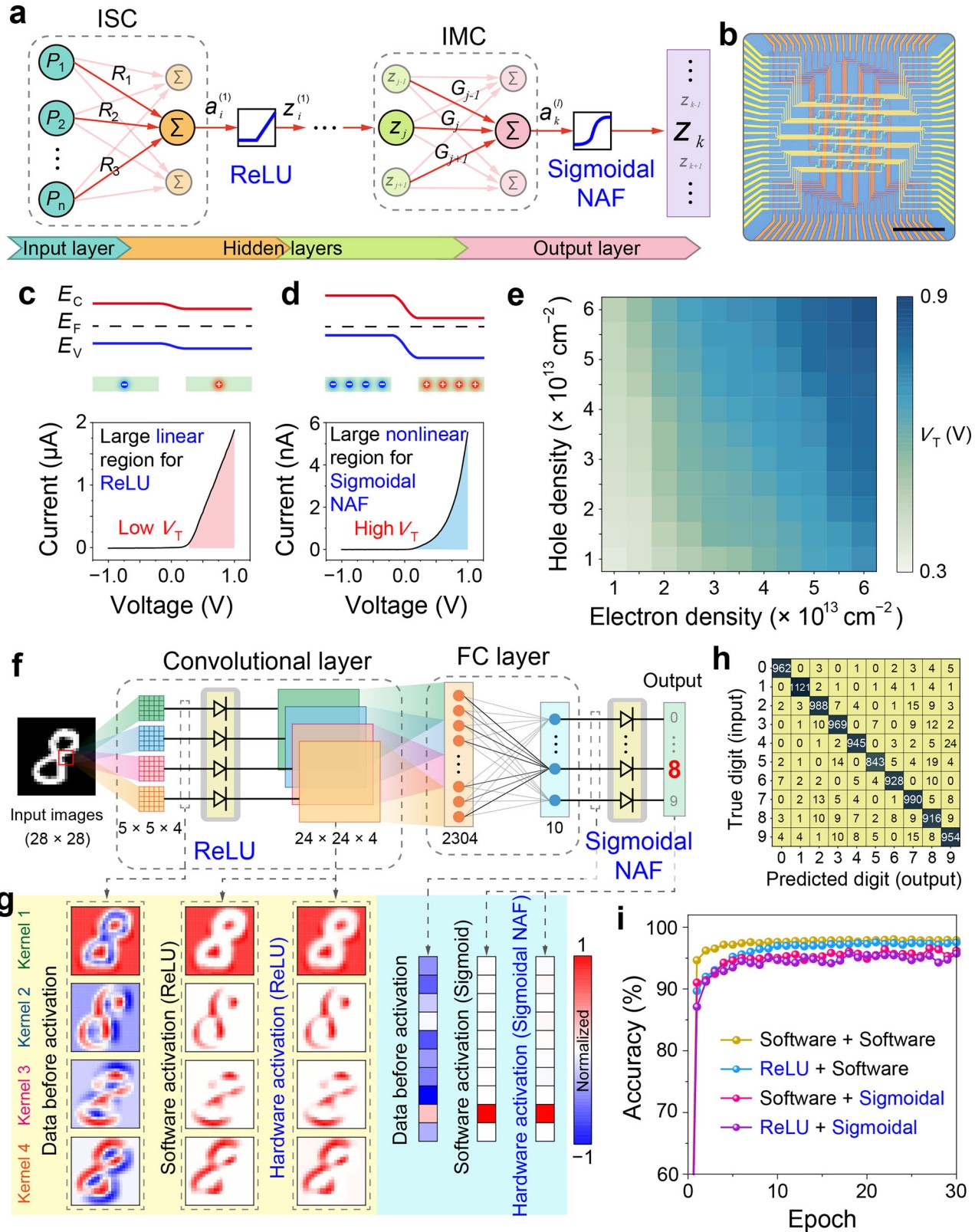

relationship across all the 63 conductance states, facilitating MVM implementation. To further validate the accuracy of analog-analog multiplication for IMC, the memory was programmed into 7 equally-divided conductance (1–7 μS) and multiplied by 10 input voltages (0.1–1.0 V) in a cross-product manner. The experimental results, shown in Fig. 3g, exhibit a small deviation from the ideal results (standard deviation of 0.48 μA, Fig. 3h). Additional 1000 analog-analog

multiplication experiments were performed using randomly selected input voltages and conductance states, with the hardware outputs closely matching the expected values (Supplementary Fig. 37). The high operation speed (8 ns), as shown in Supplementary Fig. 38, enables a high-frequency computing in IMC. Similar results have been demonstrated for the device switching between the intrinsic undoped state and the heavily n-doped state (Supplementary Note 2 and

**Fig. 4 | Neuron mode of the MM-SFGM for NAFs. a** Schematic of a neural network incorporating both linear MAC operations and nonlinear activation functions. ISC and IMC hardware can be used to perform linear operations while neuron hardware can be employed to implement ReLU and Sigmoidal NAFs. **b** Optical image of a 5 × 5 neuron-mode MM-SFGM array for nonlinear activation. Scale bar, 100 μm. Band alignments of the slightly doped (**c**) and heavily doped (**d**) WSe$_2$ channel and their corresponding rectifying curves. The doping levels of the p-type and n-type regions are controlled by the concentrations of electrons and holes stored in the split floating-gates. $V_T$ refers to the voltage at which the device transitions from nonlinear to linear I–V behavior. **e** Distribution of $V_T$ for the device under different

electron and hole concentration conditions, extracted from Supplementary Fig. 41. **f** Schematic of a CNN consisting of one conventional layer (5 × 5 × 4) and one FC layer (2304 × 10). The outputs of the four conventional kernels and the output layer are processed by the ReLU-type and Sigmoidal-type hardware neurons, respectively. **g** Output results of the hardware neurons for ReLU activation in the conventional layer and Sigmoidal activation in the output layer. **h** Confusion matrix for the MNIST dataset obtained by the CNN with hardware neurons. **i** Recognition accuracies of the CNN for the MNIST dataset. Four different combinations of software and hardware implementations for ReLU and Sigmoidal NAFs were employed in the CNN.

Supplementary Fig. 39). All these characteristics highlight the great potential of the MM-SFGMs for implementing IMC.

In addition to supporting analog computing, the MM-SFGM also enables high-speed, non-volatile data storage through appropriate configuration of its conductance switching pathways (Supplementary Note 2). In particular, when the device transitions between states such as n-p and p-p (or n-n), it exhibits a high conductance modulation ratio (~$10^7$), excellent retention characteristics (>$10^5$ s), robust cyclic endurance (>$10^6$ cycles), and fast switching speed (20 ns) (Supplementary Fig. 23). These characteristics reflect the intrinsic tunability of the device, allowing it to fulfill either computational or storage roles based on task-specific requirements. This versatility, achieved within a unified material and device framework, highlights its potential for constructing multifunctional, reconfigurable IMC architectures.

### Neuron mode of the memory for nonlinear activations
In a complete neural network, not only linear MAC and MVM operations are required, but nonlinear activation operations are also essential. Typically, the hidden layers in a neural network use ReLU activation, while the output layer utilizes NAFs like Sigmoid or Softmax (Fig. 4a). By exploiting the tunable rectification characteristics of the MM-SFGM in the p-n state, it can simulate both ReLU and Sigmoidal NAFs. As shown in Supplementary Fig. 40, the impact of electron and hole densities in the split floating-gates on the rectifying behavior of the device was systematically studied. As discussed in Supplementary Note 3, when the concentrations of electrons and holes in the floating-gates are low, the built-in potential of p-n junction is reduced, and the rectified I–V curve shows a large linear region (Fig. 4c), which is advantageous for ReLU operation. In contrast, when the electron and hole concentrations are high, the built-in potential increases, and the nonlinear region dominates in the I–V curve (Fig. 4d), making it suitable for Sigmoidal activation. Figure 4e shows the distribution of $V_T$ for the device under different electron and hole concentration conditions, where $V_T$ represents the voltage at which the device transitions from nonlinear to linear I–V behavior. A smaller $V_T$ results in a larger linear region, making it more suitable for ReLU operations. Conversely, a larger $V_T$ leads to a larger nonlinear region, making it more suitable for Sigmoidal NAF operations. As shown in Supplementary Fig. 45, the device also demonstrated nanosecond-scale operation speeds in neuron mode, performing both ReLU and Sigmoidal NAF operations with high efficiency.

To verify the ability of our device to perform nonlinear activation in neural networks, a convolutional neural network (CNN), as illustrated in Fig. 4f, was constructed. In this CNN, the output data from both the convolutional layer and the fully-connected (FC) layer were nonlinearly activated by the neuron-mode MM-SFGM hardware, as shown in Fig. 4b and Supplementary Fig. 46. The consistency of the devices' performance for both ReLU and Sigmoidal operations is demonstrated in Supplementary Figs. 66 and 67. Figure 4g and Supplementary Fig. 48 show the output results of the hardware neurons for ReLU activation in the conventional layer and Sigmoidal activation in the output layer, with performance closely matching the software simulation results. Figure 4h and i display the recognition results of the CNN using hardware neurons for the MNIST dataset, achieving a high

accuracy of 95.2%, which is comparable to the accuracy of the fully software-simulated model (97.9%).

### Reconfigurable device arrays for neuromorphic machine vision
The reconfigurable multifunctionalities of the MM-SFGM enable the construction of a NMVS integrating visual information sensing, computing, and NAFs. To construct a fully hardware neural networks involving ISC, IMC, and NAFs, here three 5 × 5 × 4 device arrays were fabricated to perform ISC, IMC, and NAFs (Fig. 5a and Supplementary Figs. 20 and 46). The performance consistency of these as-fabricated devices is demonstrated in Supplementary Note 4. Figure 5b and illustrates the structure of a fully hardware NMVS based on the MM-SFGM arrays (details in Supplementary Note 5). In this system, sensor-mode and synapse-mode device arrays are used to constitute 25 × 4 and 4 × 25 ANNs to implement ISC and IMC, respectively. Additionally, neuron-mode devices are employed to perform ReLU and Sigmoidal NAFs. This NMVS enables both supervised and unsupervised tasks for visual information.

As a demonstration, classification and autoencoding tasks were performed by this NMVS for a custom dataset containing four types of letters ("F", "L", "R" and "S", 5 × 5 pixels) with Gaussian noises ($\sigma = 0.3$) (Supplementary Fig. 76). Optical images were projected into the ISC hardware via the equipment as shown in Supplementary Fig. 77. For classification, the four output photocurrents from the ISC device array (Supplementary Fig. 78) are nonlinearly activated using Sigmoidal-type NAF devices (Supplementary Fig. 79). The four activated results as shown in Fig. 5c represent the probabilities of letters "F", "L", "R", and "S", respectively, where the substantially higher value of the target letter indicates the successful identification. Details for the training of the ANN is described in Methods. Supplementary Fig. 80 depict the photoresponsivity distribution of the ANN before and after 30 epochs, and the evolution of the recognition accuracy and loss during training are shown in Fig. 5d.

For the unsupervised autoencoding task, ISC array functions as the encoder and IMC array acts as the decoder. As shown in Fig. 5e, the four output photocurrents from ISC array are activated by the ReLU-type NAF devices, as the encoded data fed into the IMC array (Supplementary Fig. 72). The 25 output current signals from the IMC array (Supplementary Fig. 81) are further nonlinearly activated by the Sigmoidal NAF devices and are reconstructed into a 5 × 5-pixel image. The training process is described in Methods and Supplementary Note 5, and the weights distributions (photoresponsivity in ISC and conductance in IMC) of the NMVS before and after training are shown in Supplementary Fig. 82. Supplementary Fig. 83 depicts the reconstructed images of the trained NMVS for various noisy input images, and the output images of the NMVS remained almost unchanged for over 12 h owing to the local storage of the weights in the networks after training (Supplementary Figs. 84 and 85).

Notably, due to the non-volatile modulation of the photoresponsivity and conductance, the weights can be directly stored in the NMVS hardware after training, without any additional energy consumption. This contrasts with the volatile-device-based ISC/IMC hardware, which requires external power to maintain the weights. Specifically, for the ISC hardware, the sensor-mode

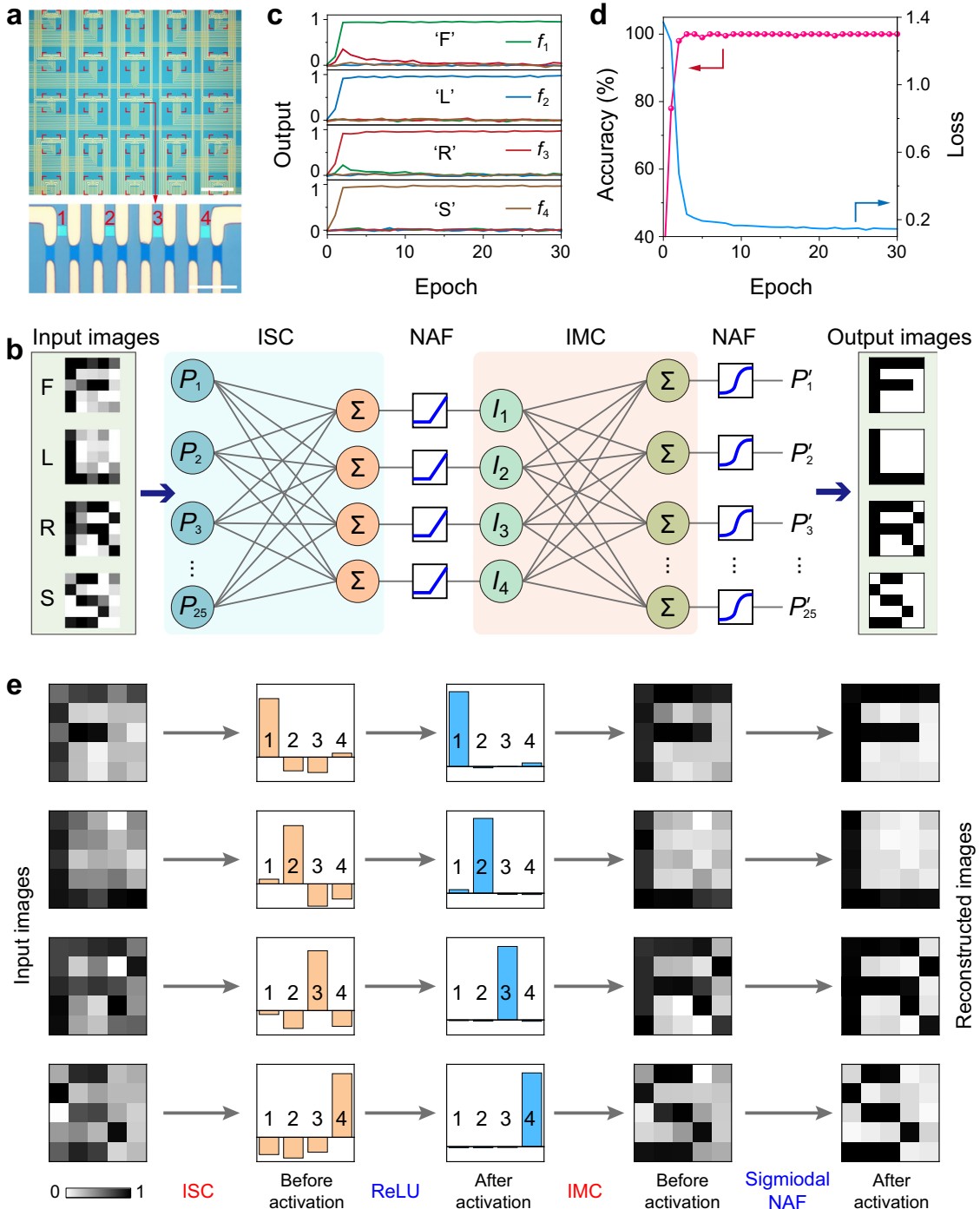

**Fig. 5 | Reconfigurable MM-SFGM arrays for NMVS. a** Optical images of a 5 × 5 × 4 MM-SFGM array. Scale bars, 300 μm and 20 μm. **b** Schematic of a NMVS consisting of a 25 × 4 ISC device array, a 4 × 25 IMC array, and ReLU/Sigmoidal-type NAF devices. **c** Average output signals ($f_1$, $f_2$, $f_3$, and $f_4$) for letters "F", "L", "R", and "S" as a function of training epochs for classification task. **d** Evolution of the classification accuracies and loss of the NMVS with training epochs. **e** Autoencoding process of 5 × 5-pixel images with Gaussian noises ($\sigma = 0.3$). The encoding process is performed by the sensor-mode device array via ISC, outputting four photocurrent signals and being activated by the ReLU-type NAF devices. The decoding process is executed by the synapse-mode device array via IMC, with the output 25 current signals being activated by the Sigmoidal-type NAF devices, thereby reconstructing 5 × 5-pixel images.

devices, which operate based on the photovoltaic effect in short-circuit current condition, sense and process optical signals in a self-powered manner. This passive current generation mechanism does not require an external power source. However, it is important to note that while the sensor array can generate output photocurrent without power consumption, the subsequent processing of the signal, such as amplification, analysis, and conversion, does require power.

## Discussion

Table 1 benchmarks the MM-SFGM against representative reconfigurable neuromorphic platforms. Most previously reported "all-in-one" devices integrate only subsets of neuromorphic primitives (e.g., ISC and IMC), or demonstrate nonlinear activation as an isolated function. In contrast, the MM-SFGM unifies ISC, IMC, and reconfigurable nonlinear activation within a single device, enabling all essential neuromorphic operations to be executed on the same physical hardware.

**Table 1 | Key parameters and functionalities of reconfigurable devices**

| Structure | Reconfigurable functions | | | | | | | | | Endurance | Scalability | Refs. |
|---|---|---|---|---|---|---|---|---|---|---|---|---|
| | Synapse mode | | | Sensor mode | | | Neuron mode | | | | | |
| | $V_{programming}$ | E | Operation Speed | $V_{programming}$ | E | Operation Speed | Mode | Operation Speed | E | | | |
| $MoTe_2$ | $V_g$ −5 ~ 5 V, $V_d$ ±20 V | 9.5 fJ | N/A | - | - | - | - | - | - | 500 | - | 67 |
| $ReSe_2$/hBN | −40 V 0.5 s 40 V 1.5 s | N/A | N/A | - | - | - | - | - | - | - | - | 68 |
| $Bi_4Ti_3O_{12}$/$MoS_2$/Graphene | ±10 V 300 ms | N/A | N/A | - | - | - | - | - | - | $5×10^5$ | - | 69 |
| $MoS_2$/$HfO_2$/$LiNbO_3$ | −1.5 V 500 μs 22 μW 7 s | N/A | N/A | - | - | - | - | - | - | 40 | 3×3 | 70 |
| α-$In_2Se_3$ | ±4 V, 0.2 s | N/A | N/A | - | - | - | - | - | - | 1000 | - | 71 |
| $TiO_2$ | ±4 V, 42 ns | 0.147 fJ | N/A | - | - | - | - | - | - | $10^4$ | - | 72 |
| $MoS_2$/hBN/Gr | ±40 V, 1 ms | N/A | N/A | - | - | - | - | - | - | 300 | - | 73 |
| BP/$Al_2O_3$/$HfO_2$/$Al_2O_3$ | −4 ~12 V 20 ms | N/A | N/A | ±18 V, 20 ms | N/A | N/A | - | - | - | 200 | 3×4 | 74 |
| PZT/SRO/STO | ±3 V, 0.15 ms | N/A | 2.6 μs | −1.94 V, 10 μs, +1.89 V, 10 μs | 3.1 nJ | 1 ns | - | - | - | $10^6$ | 3×3 | 15 |
| $MoS_2$ | ±10 V, 10 s | N/A | N/A | ±10 V, 10 s | N/A | N/A | - | - | - | - | 3×3 | 75 |
| $WSe_2$/$Al_2O_3$/$HfO_2$ | −6 V, 5 ms +4 V, 5 ms | N/A | N/A | ±6 V, 5 μs | N/A | 5 μs | - | - | - | - | 3×3 | 13 |
| $CeO_2$/STO | ±10 V, 0.5 s | N/A | N/A | ±10 V, 0.5 s | N/A | 12 ns | - | - | - | - | 3×3 | 76 |
| PZT/SRO | ±3.5 V, 0.5 μs | 1.77 nJ | 10 μs | ±10 V, 100 ns | 56 fJ | 30 μs | - | - | - | $10^9$ | 2×2 | 77 |
| Gr/3R-$WS_2$/Gr | ±2 V, 0.5 ms | N/A | N/A | ±0.05 V, 1 s | N/A | 19.2 μs | - | - | - | 2000 | 3×3×3 | 78 |
| SWNT@GDY | ±0.5 V, 1 ms | 50 aJ | N/A | ±0.5 V, 1 ms | 50 aJ | 5 ms | - | - | - | $10^5$ | 3×3×3 | 12 |
| Gr/CIPS/Gr | ±1.5 V, 0.1 s | N/A | N/A | ±0.5 V, 0.1 s | N/A | N/A | - | - | - | $10^5$ | 10×10 | 79 |
| $MoTe_2$/P(VDF-TrFE) | ±39 V, 50 μs | N/A | N/A | ±25 V, 10 μs | N/A | 1.9 μs | - | - | - | $10^6$ | 3×3×3 | 3 |
| PEDOT:$PSS^a$/perovskite/C60/bathocuproine | ±4 V, 10 s | N/A | N/A | ±4 V, 10 s | 3.63 fJ | 2.75 ns | - | - | - | - | 64×64 | 8 |
| $TiO_X$/ZnO | ±3.5 V, 100 ns | 2 nJ | N/A | ±3.5 V, 100 ns | 2 nJ | N/A | - | - | - | $10^9$ | 90×8 | 80 |
| $MoS_2$/BN/Se | ±30 V, 10 ms | N/A | N/A | ±30 V, 10 ms | N/A | N/A | - | - | - | 3000 | 3×3 | 81 |
| $VO_2$ | 3 V, 1 μs | 3.9 nJ | 30 ns | - | - | - | - | - | - | $10^{12}$ | 15×15 | 82 |
| HfZrO | +3.8 V, 100 μs −2 V, 100 μs | N/A | N/A | - | - | - | - | - | - | $2×10^7$ | - | 83 |
| $Hf_{0.5}Zr_{0.5}O_2$/$SiO_2$ | +5 V, 500 ps −3.5 V, 500 ps | 0.12 fJ | N/A | - | - | - | - | - | - | $10^7$ | - | 84 |
| Si/$HfSe_2$ | ±1 V, 1 ns | N/A | 60 ns | - | - | - | - | - | - | $2.65×10^4$ | 32×32 | 24 |
| $TiO_X$ | 4.5 V, 10 μs | N/A | 100 μs | - | - | - | - | - | - | $5×10^6$ | 20×20 | 85 |
| BP/$Al_2O_3$/$WSe_2$/h-BN | ±14 V | N/A | N/A | −6 V +5 V Persistent voltage pulse | N/A | 100 μs 5 ms | - | - | - | 55 | 3×3 | 4 |
| $VO_2$ | - | - | - | - | - | - | ReLU | 65 ns | 199.5 pJ | 5000 | - | 44 |
| μ-Si multimodal transistor | - | - | - | - | - | - | ReLU | N/A | N/A | - | - | 86 |
| $TaO_X$ memristor circuit | - | - | - | - | - | - | ReLU, Sigmoid, Tanh | N/A | N/A | - | - | 87 |
| p(V3D3-co-VI)/$Al_2O_3$ neural circuit | - | - | - | - | - | - | ReLU | N/A | 0.5 pJ | - | - | 88 |
| $NbO_2$ neural circuit | - | - | - | - | - | - | LIF | 500 μs | 1.65 pJ | 5000 | - | 89 |

**Table 1 (continued) | Key parameters and functionalities of reconfigurable devices**

| Structure | Reconfigurable functions | | | | | | | | | Endurance | Scalability | Refs. |
|---|---|---|---|---|---|---|---|---|---|---|---|---|
| | Synapse mode | | | Sensor mode | | | Neuron mode | | | | | |
| | $V_{programming}$ | E | Operation Speed | $V_{programming}$ | E | Operation Speed | Mode | Operation Speed | E | | | |
| NbO$_2$ neural circuit | - | - | - | - | - | - | LIF | - | 300 µs | N/A | - | - | 90 |
| In$_2$O$_3$ BHJ-NbO$_x$ | - | - | - | - | - | - | LIF | - | 110 ns | 1.8 nJ | 100 | 5 × 5 | 91 |
| MoS$_2$ neural circuit | - | - | - | - | - | - | IF | - | 500 ms | 2.8 nJ | N/A | 3 × 3 | 92 |
| Nb/NbTi$_x$O$_y$ neural circuit | - | - | - | - | - | - | IF | - | 200 µs | 2.4 mW | 3×10$^8$ | 3 × 2 | 93 |
| WSe$_2$/HfO$_2$/Gr/GDYO/Gr | ±1.2 V 20 ns | 4.8 fJ | 8 ns | ±1.2 V 20 ns | 4.8 fJ | 440 ns | ReLU Sigmoid | 8 ns | 16 fJ for ReLU 80 aJ for Sigmoid | 10$^6$ | 5 × 5 × 4 | **This work** |

V Voltage pulses for programming, E Energy consumption for programming, N/A Not available. "-": No such feature.
ᵃPEDOT PSS refers to poly(3,4-ethylenedioxythiophene):poly(styrene sulfonate).

Beyond functional completeness, this trifunctional integration directly addresses the system-level compactness and energy-efficiency challenges motivating this work. In conventional neuromorphic architectures, nonlinear activation is typically implemented using external digital processors or analog CMOS circuits[3,10,12,50], which introduce substantial peripheral overhead. As summarized in Table 2, reported activation devices and circuits generally occupy large areas and consume µW–mW-level power, often dominating the system footprint and energy budget even when synaptic elements are efficient. By contrast, the MM-SFGM realizes ReLU- and Sigmoid-type activation natively at the device level, without auxiliary circuitry. The nonlinear activation is achieved through electrically reconfigurable rectification within the same transistor used for sensing or memory operations, resulting in femtojoule- to attojoule-level activation energy and a compact device footprint (Table 1). It is important to note that the reported device area is obtained using laboratory-scale fabrication processes. With higher-resolution nanolithography and advanced integration techniques, the device footprint can be further reduced by orders of magnitude, indicating substantial headroom for future scaling. At the system level, consolidating sensing, linear computing, and nonlinear activation into a single device eliminates repeated data movement between separate functional blocks, reducing latency and energy consumption across the inference pipeline. As demonstrated in the hardware neuromorphic vision system, all processing stages—including convolution, vector–matrix multiplication, and nonlinear activation—are executed within MM-SFGM arrays, achieving nanosecond-scale response and femtojoule-level energy consumption per operation. Although the current demonstration employs a 100-device array and laboratory-scale instrumentation, the compactness emphasized here refers to functional density at the device and array level, rather than the experimental setup. The MM-SFGM architecture is inherently scalable, and continued advances in wafer-scale 2D material synthesis[60–63] and monolithic integration techniques[64–66] are expected to enable larger and denser arrays. While challenges such as interconnect resistance, sneak paths, and defect tolerance remain, the present results establish a clear architectural pathway toward compact and energy-efficient neuromorphic hardware.

In conclusion, we propose a high-speed, reconfigurable MM-SFGM that integrates sensing, memory, computation, and nonlinear activation within a single compact hardware unit and material platform. Through array-level implementation and full-system demonstration, we establish a qualitative milestone in monolithic neuromorphic hardware capable of performing all fundamental operations required for intelligent sensing and processing. This architecture provides a general-purpose, energy-efficient foundation for next-generation edge-AI and multifunctional computing systems, offering a direct pathway to bridge device-level physical mechanisms with system-level intelligence, while circumventing the limitations of conventional multi-device silicon architectures.

## Methods

### Device array fabrication

The fabrication process of the 5 × 5 × 4 MM-SFGM array is illustrated in Supplementary Fig. 2. First, chemical-vapor-deposition (CVD) grown MLG with a thickness of approximately 1.4 nm was transferred onto a Si/ SiO$_2$ (285 nm) substrate and patterned to square array by the standard photolithography and oxygen plasma etching. Cr/Au (10/ 50 nm) control-gate electrodes (CG1/CG2) were deposited on the MLG via electron beam evaporation. Next, prepatterned GDYO film (~9.8 nm, Supplementary Fig. 55) and MLG film (~2.1 nm) were transferred and stacked on top of the bottom MLG, forming the MLG/ GDYO/MLG heterostructure. The precise alignment and stacking of the prepatterned 2D material arrays were performed using a high-precision transfer and stacking platform (WeDo Tech., Tianjin). The heterostructure was then split to two pieces using focused ion beam

**Table 2 | Comparison of key parameters of nonlinear activation devices and circuits**

| Function | Type | Energy (power) consumption | Area | Delay time | Refs. |
|---|---|---|---|---|---|
| Sigmoid, Hyperbolic tangent, Soft-sign, Softplus, ELU, SeLU | CMOS | 3.7 mW (5 bit), 4.6 mW (8 bit) | 0.003 mm²@16 nm | N/A | 28 |
| Tanh | CMOS | 0.071 mW | 582 μm²@90 nm | 1.40 ns | 29 |
| ReLU, Leaky ReLU, Tanh | CMOS | 2.155 mW | 35.809 μm²@180 nm | 420 ps | 30 |
| Sigmoid | CMOS | 11.4 μW | 184 μm²@180 nm | N/A | 31 |
| Tanh, Sigmoid | CMOS | 0.07 mW | 1554 μm²@180 nm | 2.25 ns | 32 |
| ReLU, Sigmoid | CMOS | 55-88 mW | 0.028 mm²@45 nm | 570 ps | 35 |
| ReLU | Mott | 199.5 pJ | 0.64 μm² | 61.4 ns | 44 |
| ReLU | ADC | 29.6 fJ | 45 μm² | N/A | 37 |
| ReLU, Tanh, Sigmoid | ADC | 4.75 nJ | 0.086 mm²@90 nm | 207 ns | 38 |
| ReLU | ADC | 87 μW | 7200 μm²@180 nm | 2.1 μs | 39 |
| Tanh, clipped-ReLU | ADC | 15 μW | 101.64 μm²@28 nm | N/A | 41 |
| ReLU | ADC | 3.08 TOPS/W | 3.91 TOPS/mm²@28 nm | N/A | 42 |
| Mish, GELU, SiLU | ADC | 168 μW | 1218 μm²@65 nm | N/A | 43 |
| Tanh, ReLU | Digital processors | 1982 mJ | 263 mm² | N/A | 94 |
| Tanh | ASIC | N/A | 17864.24 μm²@180 nm | 2.45 ns | 95 |
| Sigmoid Tanh | ASIC | 2.16 mW 2.64 mW | 7973.2 μm²@65 nm 10180 μm²@65 nm | 3 ns, 3.16 ns | 96 |
| Sigmoid | ASIC | 2.66 mW | 8097.48 μm²@90 nm | 1.96 ns | 97 |
| ReLU | ASIC | 20.4 mW | 130 nm | 14 ns | 98 |
| Sigmoid | FPGA | 19 mW | N/A | 55 ns | 99 |
| Tanh | FPGA | N/A | N/A | 45.4 ns | 100 |
| Sigmoid, Tanh | ASIC FPGA | 0.31 mW, 0.28 mW 56 mW, 56 mW | 4678.78 μm²@180 nm 5582.5 μm² | 1.6 ns, 1.7 ns 23.1 ns, 23 ns | 101 |
| Sigmoid | MRAM | 18.04 μW | 0.138 μm² | 48 ns | 102 |
| ReLU, Sigmoid | Memory | 16 fJ for ReLU 80 aJ for Sigmoid | 15 μm² | <8 ns | This work |

"@* nm" represents node process technology.

*N/A* Not available.

(FIB) with a gap of approximately 110 nm (Supplementary Fig. 4). The top MLG layer was pretreated with UV-ozone for 10 s to introduce surface dangling bonds, which facilitated the deposition of a 10.5 nm $HfO_2$ dielectric layer on the heterostructure via atomic layer deposition (ALD, Supplementary Fig. 6). Subsequently, CVD-grown multilayer $WSe_2$ (~5.6 nm, Supplementary Fig. 53) channels were stacked on $HfO_2$, and the source and drain electrodes (Cr/Au 10/50 nm) were deposited. Finally, the device array was mounted and wire-bonded to a probe-card with 400 pins (Supplementary Fig. 77a) for measurement. The CVD growth was performed using Micro-STS1200 system (Units Technology, www.units-tech.com.cn). Supplementary Fig. 3 presents the corresponding atomic force microscope (AFM) images during fabrication, highlighting the clean surface of each layer.

**Measurement setup**

The basic electrical characteristics of the devices were measured by a semiconductor analyzer (Keithley 4200A-SCS) equipped with a pulse measurement unit (PMU). An arbitrary waveform generator (Tektronix, AFG31052) was used to generate nanosecond gate pulses for the reconfiguration of the devices and updating of the weights. The photocurrent maps generated by the p-n and n-p homojunctions (Fig. 2a) were measured using a WITec (Alpha 300RAS) photocurrent testing module. Prior to testing, the MM-SFGM was programmed to the p-n or n-p states and placed on the testing platform with both CG1 and CG2 floating. A 532 nm laser with an intensity of 100 nW was focused through a 100× objective lens onto the $WSe_2$ channel. The source and drain electrodes were connected to a transimpedance amplifier (TIA, DHPCA-100) to measure the photocurrents generated at each position. The

photocurrent map (100 × 100 pixels) was generated by moving the device with a piezoelectric platform, with movement and measurement controlled automatically by the WITec system. For the ISC, 5 × 5-pixel images were generated by a 532 nm laser and a spatial light modulator (SLM) operating in intensity-modulation mode. Then the input images were adjusted to 1 mm × 1 mm via optical lenses, and projected to the 5 × 5 × 4 sensor array (Supplementary Fig. 77).

**Implementation of image classification via NMVS**

As illustrated in Supplementary Fig. 79, the NMVS for classification task consists of a 5 × 5 × 4 sensor-mode device array for ISC and four neuron-mode devices for NAF. The 5 × 5 × 4 device array is divided into 5 × 5 pixels and each pixel consists of 4 subpixels. ISC operations were executed by connecting the devices with the same subpixel index ($m = 1, 2, 3, 4$) in each pixel in parallel and accumulating the generated photocurrents (Supplementary Fig. 78). The initial weight distribution (photoresponsivity) of the sensor array was randomly generated. For each training epoch, 5 × 5-pixel images from the training dataset were optically projected to the sensor array, and the parallelly connected circuit outputted four photocurrent signals $I_m$:

$$I_m = \sum_{i=1}^{5}\sum_{j=1}^{5} I_{m,ij} = \sum_{i=1}^{5}\sum_{j=1}^{5} P_{ij} \cdot R_{m,ij}, \qquad (1)$$

where $i$ and $j$ represent the row and column index. $I_m$ was further activated by four Sigmoidal-type NAF devices, outputting four signals ($f_1, f_2, f_3$, and $f_4$) representing the probabilities for letters "F", "L", "R",

and "S". After each epoch, the gradients of loss function L (here cross-entropy function) was backpropagated to update the weights of the neural network:

$$\Delta R_{m,ij} = \frac{\eta}{S} \sum_{k=1}^{S} \nabla_{R_{m,ij}} L_k, \qquad (2)$$

$$L = -\sum_{k=1}^{M} y_k \log \hat{y}_k, \qquad (3)$$

where $y_k$ and $\hat{y}_k$ were the label and prediction, respectively, and learning rate $\eta = 0.2$ and the size of batch $S = 20$. Backpropagation was executed by the FPGA integrated on the same PCB (Supplementary Fig. 71). Based on computed gradients, the FPGA and MUX array coordinated voltage pulses to selectively update the photo-responsivity of sensor-mode devices after each epoch, enabling efficient in-situ learning (Supplementary Note 5.1). Classification accuracy was evaluated by projecting test images during each epoch.

### Implementation of image autoencoding via NMVS

The structure of the NMVS for autoencoding task is shown in Fig. 5b and Supplementary Figs. 71 and 72, which consists of a 25 × 4 ANN for ISC, a 4 × 25 ANN for IMC, 4 ReLU-type and 25 Sigmoidal-type NAF devices. The initial weights (photoresponsivity for ISC and conductance for IMC) of the ANNs were randomly generated. For each training epoch, 5 × 5-pixel noisy images chosen from the training dataset were optically projected onto the sensor array, outputting four photocurrents $I_m$. A ReLU operation was performed in the following via the four ReLU-type NAF devices, and the output current signals were converted to voltage and fed into the 4 × 25 IMC array (Supplementary Fig. 81). The 25 output signals were further nonlinearly activated by the 25 Sigmoidal-type NAF devices, which were used to reconstruct 5 × 5-pixel images. The loss function (mean-square error, $L = \sum_{i=1}^{5} \sum_{j=1}^{5} (p_{i,j} - p'_{i,j})^2$, where $P_{i,j}$ and $P'_{i,j}$ correspond to the intensity of the ($i$, $j$) pixel in the input and reconstructed images, respectively) was backpropagated via the on-board FPGA to compute gradients. Based on these results, the FPGA and MUX array coordinated voltage pulses to update the photo-responsivity and conductance of ISC and IMC devices, respectively, after each training epoch (Supplementary Note 5.1).

## Data availability

The data generated in this study have been deposited in Figshare at https://figshare.com/articles/dataset/Source_data_for_NCOMMS-25-48768-T/30325393.

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

## Acknowledgements

This work was supported by the National Key Research and Development Program of China (2021YFA1400601 (J.T.)), the National Natural Science Foundation of China (62274119 (X.C.)), Basic Research Development Program of SuZhou (SJC2023004 (L.K.)), and the Shenzhen Science and Technology Program (JCYJ20240813160206009 (Z.C.)), and the Fundamental Research Funds for the Central Universities (010-63253118 (X.C.)). The authors are grateful for the technical support for Nano-X from Suzhou Institute of Nano-Tech and Nano-Bionics, Chinese Academy of Sciences (SINANO).

## Author contributions

Z.Z., L.K., and X.C. designed the project, and X.C., J.T., and T.L. supervised this project. Z.Z., Y.L., and J.Y. fabricated the device and performed the electrical/optoelectronic measurements. F.W. prepared the GDYO film. S.S. and H.Q. performed the characterization. X.C. wrote the manuscript, Z.L. and Z.C. revised the manuscript. All the authors discussed the results and commented on the manuscript.

## Competing interests

The authors declare no competing interests.
