## [Transparent Peer Review file · Nature Communications]

A reconfigurable photosensitive split-floating-gate memory for neuromorphic computing and nonlinear activation

Corresponding Author: Dr Xu-Dong Chen

Version 0:

Reviewer comments:

Reviewer #1

(Remarks to the Author)

The paper is generally good with me. Well done folks. Here is another potential improvement.

In Fig. 3e, the plot seems to saturate above a conductance of $1e-7$ S. That's interesting because it means that your errors do not grow even if you increase your conductance, unlike many other forms of storage. I suggest you add a few pieces of data from the literature to this plot (using those papers that have reported similar conductance vs. error plots). I suspect that most of them would lie on one side of your plot (likely with higher errors) and also exhibit a lower range compared to you (in terms of conductance range). See these papers in particular: <https://doi.org/10.1038/s41586-022-04992-8>, <https://doi.org/10.1109/IEDM19574.2021.9720597>, <https://doi.org/10.1038/s41467-020-16108-9>, <https://doi.org/10.1126/science.adi9405>

IF your data does beat most/all of these past reports, then consider making it a much larger plot and expand its prominence in the figure as well as the text.

Besides, I believe that Reviewer 2's comments are highly valid, but most of them have been addressed by the authors in obscure corners of the paper (especially hidden in the long and hard-to-read supplement). As such, Reviewer 2's problems would also be problems most average readers would face. I will advise the authors on how to fix the specific issues raised by Reviewer 2, which I hope will also help the editor make a decision.

1. The need for the special graphdiyne oxide layer (compared to regular bulk flash layers) is elaborated in Supplementary Sections 1.2 - 1.4. However, the main text does not clearly say why the device needs this special structure. Please add a few sentences in the mechanism section of the main text briefly stating why the graphdiyne oxide is needed to enable the new functions (e.g., it undergoes TS, it enables high-speed switching, etc.), and why a traditional bulk flash structure would not work.

2.1. Supplementary Table 4 compares reconfigurable devices in the literature and shows that only this work offers all the functions in a single device. I would suggest you add a few more neuron devices (e.g., NbO₂) to provide balance. Reviewer 2 is asking for "benchmarks" for functional integration - Table 4 is a qualitative benchmark showing that this level of functional integration (with these functions) has not been done before. Please add some more sentences to the main text referencing Table 4 and saying that you have offered a comparison to past literature.

2.2. "Heterogeneous integration" is just bad phrasing. The authors are trying to say that multi-functional chips require combining of many devices, each expressing a different function. Instead, this work offers a single device that combines many of these functions. This device is built by carefully integrating many materials with specific active properties. "Heterogeneous integration" in the materials science literature usually means combining many materials in a single chip/device, which is what you are doing. I suggest you just drop that phrase and use descriptive sentences instead.

2.3. Reviewer 2 is correct in identifying that this paper does not demonstrate quantitative superiority in performance compared to the best processors in the market (or in the research space). The main claim in this paper is the first qualitative

combination of new functions, and not quantitative superiority. While Supplementary Table 4 does show some quantitative improvements compared to past devices, and Fig 3e may end up being quantitatively better than many published works, please don't make any strong claims around quantitative performance. You would need a lot of engineering improvements to get performance that can beat any existing research or commercial devices. "Qualitatively first demonstrations" are extremely important in research (even if there is no indication of quantitative superiority) - quantitative advances will usually follow qualitative discoveries. As for publication, this issue is not prohibitive, and I have seen many well received/cited papers by this publisher on similar qualitative discoveries. Authors: please ensure that you clearly state that this is the first qualitative demo of all these functions, and quantitative improvements need a lot more engineering efforts in the future.

Reviewer #2

(Remarks to the Author)

The manuscript "A reconfigurable photosensitive split-floating-gate memory for in-memory sensing, computing, and nonlinear activation" reports a split-floating-gate WSe₂ memory with graphdiyne oxide as a threshold-switching layer, which has multi-functions of in-sensor computing, in-memory computing, and nonlinear activation functions. The authors conducted a very detailed study on the performance of the device and demonstrated the scalability of the large-scale array. However, the reviewer does not recommend this manuscript for publication in Nature Communications before the innovation and the significance of the key scientific issues have been further improved. The detailed reasons are as follows.

1. The innovation of the device structure is unclear. In the manuscript, a special memory structure was adopted: the blocking layer in the traditional flash storage medium was replaced with graphdiyne oxide. What is the necessity of adopting this particular structure? In my opinion, the traditional storage stack can also achieve multi-functional demonstrations.
2. The functionality is not innovative enough. In the manuscript, the key scientific issues are described as "However, existing post-silicon and neuromorphic systems often implement only a subset of these functions and rely heavily on heterogeneous integration, increasing system complexity and limiting scalability."
 - 1) There have been numerous previous studies on all-in-one prototype devices and array-level research. This manuscript does not present any benchmarks to demonstrate the innovative aspects of this work in terms of functional integration.
 - 2) The multi-functional integration heavily relies on heterogeneous integration. This work did not solve this problem, and it is also based on multi-layer heterogeneous integration.
 3. The manuscript does not provide a clear description of the quantitative breakthroughs in key performance indicators.

Version 1:

Reviewer comments:

Reviewer #1

(Remarks to the Author)

Editor, I'm good with the paper, and I think peer review has done its job. I recommend you send it to print, and I don't need to see it again.

I'm impressed by Fig. 3e. I suggest you make it larger. Optionally, I also suggest you add a couple of additional data sets to it, especially from Fig. 3c of this paper: <https://arxiv.org/abs/2505.15936>. You will notice that your data offers lower conductances for a given error/standard deviation compared to the paper I referenced (meaning that you offer lower power consumption for a given error), as well as extending the range of the data from the referenced paper. In addition, your data offers both lower conductances as well as lower errors compared to most other published literature.

Suhas Kumar

Reviewer #2

(Remarks to the Author)

I think the authors has provided a large amount of experimental data, which proves that a great deal of time and effort were invested in this work. However, the biggest obstacle hindering the development of this article lies in the appropriate organization. And I still believe that this work needs to be major revised and the article organization needs to be re-done before it can be published.

My previous objection to this work's publication on Nature Communications was related to two insufficient points: (1) the novelty of the research; (2) whether the key scientific issues identified by the author as needing to be addressed in the research have been successfully resolved through innovation.

In the author's response, it has been clearly stated that the focus of the research lies in the fact that these three functions (in-sensor computing, in-memory computing and multiple nonlinear activation functions) can be achieved on a single device, rather than in the pursuit of a breakthrough in a specific single performance. The key scientific question that the author aims to address is existing works not compact, energy-efficient enough. So, what was the final result? How compact and energy-efficient? The images captured by the mm-scale optical microscope are not compact, and there is no discussion regarding energy efficiency. I totally agree with Reviewer 1 that not all tasks require quantification, but even for the qualitative study on "How compact and energy-efficient", there is no such discussion in this article on these core issues. Among the 5 figures in the manuscript, 1 figure is about the innovation of the device structure, 3 figures are showing the implementation of a single function, and the last figure only show the demonstration of function integration but no comparison. None about "How compact and energy-efficient". This led to my previous question, which is that the innovation of the device structure and

single performance have not been adequately demonstrated to support the publication of the paper. And it also confused me that although traditional split-gate flash (reported by a large amount of works) can also achieve these integration functions, they simply haven't conducted these tests.

In conclusion, I believe it is necessary to clearly define the core issues that the research needs to address and provide sufficient data support or discussion on the core issues.

Supplementary Tables 3 and 4 present comparisons of energy consumption and area efficiency. Although this work does not achieve the best performance among the compared studies, it still remains at a leading level. Additionally, my original suggestion was for the authors to highlight these comparisons in the main text, as they represent the innovative aspects emphasized in the title of their work. Therefore, while the data provided in the Supplementary Information (SI) can meet the requirements of the question, it would be better to include these comparisons in the main text to better reflect these aspects.

Reviewer #1 (Remarks to the Author):

The paper is generally good with me. Well done folks. Here is another potential improvement.

In Fig. 3e, the plot seems to saturate above a conductance of $1e-7$ S. That's interesting because it means that your errors do not grow even if you increase your conductance, unlike many other forms of storage. I suggest you add a few pieces of data from the literature to this plot (using those papers that have reported similar conductance vs. error plots). I suspect that most of them would lie on one side of your plot (likely with higher errors) and also exhibit a lower range compared to you (in terms of conductance range). See these papers in particular: <https://doi.org/10.1038/s41586-022-04992-8>, <https://doi.org/10.1109/IEDM19574.2021.9720597>, <https://doi.org/10.1038/s41467-020-16108-9>, <https://doi.org/10.1126/science.adi9405>.

If your data does beat most/all of these past reports, then consider making it a much larger plot and expand its prominence in the figure as well as the text.

Reply: We sincerely thank you for your positive assessment and for the insightful suggestion regarding Fig. 3e. Following your recommendation, we have incorporated benchmark data from six representative literature reports (Refs. 55–60) that provide comparable conductance–error characteristics. The updated Fig. 3e and related discussion (Pages 10–11, Lines 223–236) now emphasize both the conductance range and the stability performance of our device relative to prior works.

(1) Conductance range and stability analysis.

Figure 3e presents the standard deviation (SD) of the read conductance as a function of the programmed conductance across a wide range from 10 pS to 7 μ S. By repeatedly programming and reading 30 distinct conductance states, we find that the SD remains below 13 nS throughout this range. In the low-conductance region (10 pS–0.1 μ S), SD increases with conductance, reflecting predictable and controllable noise behavior that enables precise tuning of small weights. In the high-conductance region (0.1 μ S–7 μ S), SD saturates and no longer increases with conductance, confirming that higher conductance does not deteriorate device stability. This saturation behavior indicates that our device maintains low noise and high reliability even under large-weight conditions, which is essential for accurate inference in neuromorphic computation.

(2) Comparison with previous studies.

To benchmark our results, we added the conductance–SD data from six key literature sources (Refs. 55–60) to Fig. 3e. For consistency, all datasets were converted to logarithmic scales. The comparison reveals that previous devices typically exhibit a narrower conductance window (often spanning only 1–2 orders of magnitude, usually

1 μS –1 mS) and significantly higher SD values (0.1–10 μS). In contrast, our MM-SFGM demonstrates an ultra-wide dynamic range (10 pS–7 μS) with SD < 13 nS throughout, indicating both broader tunability and superior noise suppression. This highlights the unique capability of our device to maintain high precision and stability across multiple operation states, surpassing previously reported results.

These advantages make our MM-SFGM a promising building block for high-precision, energy-efficient, and large-scale neuromorphic and AI accelerator systems.

Fig. 3e, Standard deviation (SD) of conductance as a function of conductance, measured over a range from 10 pS to 7 μS . Each conductance state was programmed ten times and read one hundred times after each write, with the SD calculated from the distribution of read values. Representative data from previous reports are included for comparison.

Besides, I believe that Reviewer 2's comments are highly valid, but most of them have been addressed by the authors in obscure corners of the paper (especially hidden in the long and hard-to-read supplement). As such, Reviewer 2's problems would also be problems most average readers would face. I will advise the authors on how to fix the specific issues raised by Reviewer 2, which I hope will also help the editor make a decision.

1. The need for the special graphdiyne oxide layer (compared to regular bulk flash layers) is elaborated in Supplementary Sections 1.2 - 1.4. However, the main text does not clearly say why the device needs this special structure. Please add a few sentences in the mechanism section of the main text briefly stating why the graphdiyne oxide is needed to enable the new functions (e.g., it undergoes TS, it enables high-speed switching, etc.), and why a traditional bulk flash structure would not work.

Reply: We sincerely thank you for this very helpful suggestion. You are absolutely right that, in the original manuscript, the necessity of the graphdiyne oxide (GDYO) layer

was only discussed in detail in Supplementary Sections 1.2–1.4, which could make it less accessible to readers.

Following your advice, we have now added explicit clarification in the “Mechanism of the MM-SFGM with reconfigurable multifunctionality” section of the main text (Pages 3–4, Lines 85–96). In the revised text, we briefly explain that the GDYO layer provides volatile threshold-switching (TS) characteristics, which enable ultrafast charge injection into the floating gates under low voltages and then spontaneously recover to the insulating state to suppress charge leakage. This mechanism is fundamentally different from the tunneling process in conventional bulk flash devices, which typically requires higher voltages and longer programming times, resulting in slower and less energy-efficient operation. By highlighting these contrasts, the revised manuscript now makes clear why GDYO is essential for enabling high-speed switching, low-energy operation, and multifunctional reconfigurability, whereas a traditional flash stack would not achieve the same performance.

We appreciate your constructive comment, which has significantly improved the clarity and accessibility of our work.

2.1. Supplementary Table 4 compares reconfigurable devices in the literature and shows that only this work offers all the functions in a single device. I would suggest you add a few more neuron devices (e.g., NbO₂) to provide balance. Reviewer 2 is asking for "benchmarks" for functional integration - Table 4 is a qualitative benchmark showing that this level of functional integration (with these functions) has not been done before. Please add some more sentences to the main text referencing Table 4 and saying that you have offered a comparison to past literature.

Reply: We sincerely thank you for this constructive suggestion. You are correct that, in order to provide a more balanced comparison, additional neuron-type devices should be included in Supplementary Table 4. Following your advice, we have expanded Supplementary Table 4 by adding representative neuron devices (e.g., NbO₂), which further strengthens the benchmarking against the state-of-the-art.

In addition, as you recommended, we have revised the main text (Discussion section, Page 18, Lines 397–402) to explicitly reference Supplementary Table 4 and to highlight that our MM-SFGM is, to the best of our knowledge, the only device reported so far that monolithically integrates ISC, IMC, and reconfigurable nonlinear activation functions (NAFs). This added discussion clarifies the benchmarking and makes the novelty of our functional integration more explicit for the reader.

We are grateful for your helpful feedback, which has improved both the completeness

and the clarity of our manuscript.

2.2. "Heterogeneous integration" is just bad phrasing. The authors are trying to say that multi-functional chips require combining of many devices, each expressing a different function. Instead, this work offers a single device that combines many of these functions. This device is built by carefully integrating many materials with specific active properties. "Heterogeneous integration" in the materials science literature usually means combining many materials in a single chip/device, which is what you are doing. I suggest you just drop that phrase and use descriptive sentences instead.

Reply: We sincerely thank you for this helpful clarification. As you correctly pointed out, the term “heterogeneous integration” has a specific meaning in materials science—referring to the integration of multiple materials within a single device—which indeed applies to our structure. To prevent ambiguity, we have **removed the term** from the manuscript and replaced it with clear descriptive sentences:

Abstract (Page 1, Lines 30–31):

Original: “... rely heavily on heterogeneous integration, increasing system complexity and limiting scalability.”

Revised: “... rely on assembling multiple discrete devices to realize different operations, which increases system complexity and limits scalability.”

Introduction (Page 3, Lines 75–76):

Original: “... breaks from the traditional need for heterogeneous integration, offering a compact, reconfigurable, and general-purpose solution ...”

Revised: “... eliminates the traditional need to combine multiple device types for different functions, offering a compact, reconfigurable, and general-purpose solution ...”

Discussion/Conclusion (Page 19, Lines 430–431):

Original: “... while circumventing the limitations of conventional heterogeneous silicon integration.”

Revised: “... while circumventing the limitations of conventional multi-device silicon architectures.”

2.3. Reviewer 2 is correct in identifying that this paper does not demonstrate quantitative superiority in performance compared to the best processors in the market (or in the research space). The main claim in this paper is the first qualitative

combination of new functions, and not quantitative superiority. While Supplementary Table 4 does show some quantitative improvements compared to past devices, and Fig 3e may end up being quantitatively better than many published works, please don't make any strong claims around quantitative performance. You would need a lot of engineering improvements to get performance that can beat any existing research or commercial devices. "Qualitatively first demonstrations" are extremely important in research (even if there is no indication of quantitative superiority) - quantitative advances will usually follow qualitative discoveries. As for publication, this issue is not prohibitive, and I have seen many well received/cited papers by this publisher on similar qualitative discoveries. Authors: please ensure that you clearly state that this is the first qualitative demo of all these functions, and quantitative improvements need a lot more engineering efforts in the future.

Reply: We thank you very much for your constructive feedback and for clarifying the appropriate positioning of our claims. Following your valuable suggestion, we have revised the manuscript to explicitly emphasize that this work represents the **first qualitative demonstration** of ISC, IMC, and reconfigurable NAFs being integrated in a single monolithic device, rather than a claim of quantitative performance superiority.

As you rightly pointed out, achieving quantitative advances will require substantial engineering optimization beyond the current proof-of-concept demonstration. We have accordingly modified the **Discussion** section (Pages 18–19, Lines 402, 407–413, 425–427) to clearly communicate this positioning and to highlight the qualitative nature and conceptual significance of our findings.

We would like to express our heartfelt gratitude to you for the indispensable role you have played in shaping this work. Your insightful guidance not only inspired us to fully explore the NAF functionality of our device, but also greatly elevated the originality and impact of our study. Throughout the multiple rounds of review, you patiently and thoroughly addressed our questions, offering constructive suggestions with remarkable dedication and professionalism. Your careful attention and thoughtful feedback have been invaluable to us. We sincerely thank you for your time, effort, and encouragement, and we wish you continued success and all the best in your endeavors.

Reviewer #2 (Remarks to the Author):

The manuscript “A reconfigurable photosensitive split-floating-gate memory for in-memory sensing, computing, and nonlinear activation” reports a split-floating-gate WSe₂ memory with graphdiyne oxide as a threshold-switching layer, which has multi-functions of in-sensor computing, in-memory computing, and nonlinear activation functions. The authors conducted a very detailed study on the performance of the device and demonstrated the scalability of the large-scale array. However, the reviewer does not recommend this manuscript for publication in Nature Communications before the innovation and the significance of the key scientific issues have been further improved. The detailed reasons are as follows.

We sincerely thank you for your careful and thoughtful review of our manuscript. Your detailed and constructive comments have provided valuable guidance, highlighting both the strengths of our work and areas where further improvements in innovation and significance are needed. We greatly appreciate your patience and the time you dedicated to reviewing our study, and we hope that our responses adequately address your questions while reflecting our efforts to incorporate your insightful suggestions.

1. The innovation of the device structure is unclear. In the manuscript, a special memory structure was adopted: the blocking layer in the traditional flash storage medium was replaced with graphdiyne oxide. What is the necessity of adopting this particular structure? In my opinion, the traditional storage stack can also achieve multi-functional demonstrations.

Reply: We thank you for your critical comment on the innovation of the device structure. You are correct that conventional flash memories can in principle also achieve multifunctional reconfiguration. The key reason we adopted graphdiyne oxide (GDYO) instead of a traditional blocking dielectric is that GDYO provides a volatile threshold-switching (TS) mechanism. This mechanism allows charge injection into the floating gates within nanoseconds under very low voltages, and then automatically returns to the insulating state, ensuring long-term non-volatile charge storage. In contrast, traditional flash memories achieve non-volatility via tunneling through thick insulating barriers, which requires much higher voltages and longer times, thereby increasing energy consumption and slowing down state reconfiguration.

To address your concern, we have added explicit discussion in the “Mechanism of the MM-SFGM with reconfigurable multifunctionality” section of the revised manuscript (Pages 3–4, Lines 85–96). We believe this addition makes the structural innovation and necessity of GDYO much clearer.

2. The functionality is not innovative enough. In the manuscript, the key scientific issues are described as “However, existing post-silicon and neuromorphic systems often implement only a subset of these functions and rely heavily on heterogeneous integration, increasing system complexity and limiting scalability.”

1) There have been numerous previous studies on all-in-one prototype devices and array-level research. This manuscript does not present any benchmarks to demonstrate the innovative aspects of this work in terms of functional integration.

Reply: We thank you for raising this important point. You are correct that a number of previous reports have presented “all-in-one” prototype devices. However, as we clarify in the revised manuscript, the majority of these devices only combine synaptic and sensor functions, i.e., they can perform in-memory computing (IMC) and in-sensor computing (ISC). A few other platforms have demonstrated ReLU-type activation functions, but these devices only provide activation and do not simultaneously support IMC or ISC.

As summarized in Supplementary Table 4, our MM-SFGM is, to the best of our knowledge, the **only device that monolithically integrates all three essential neuromorphic operations—ISC, IMC, and reconfigurable nonlinear activation functions (NAFs)**. This represents a significant functional expansion for reconfigurable devices, enabling a single device platform to perform multiple roles typically distributed across different elements in artificial neural networks.

To make this distinction clearer, we have revised the **Discussion** section of the main text (Page 18, Lines 397–402) to explicitly reference Supplementary Table 4 and highlight that our work uniquely achieves trifunctional integration, going beyond prior all-in-one demonstrations.

Supplementary Table 4. Key parameters and functionalities of reconfigurable devices.

Structure	Reconfigurable functions									Endurance	Scalability	Ref.
	Synapse mode			Sensor mode			Neuron mode					
	$V_{\text{programming}}$	E	Operation Speed	$V_{\text{programming}}$	E	Operation Speed	Mode	Operation Speed	E			
MoTe ₂	V_g -5~5 V, V_d \pm 20 V	9.5 fJ	N/A	-	-	-	-	-	-	500	-	41
ReSe ₂ /hBN	-40 V 0.5 s 40 V 1.5 s	N/A	N/A	-	-	-	-	-	-	-	-	42
Bi ₄ Ti ₃ O ₁₂ /MoS ₂ / Graphene	\pm 10 V 300 ms	N/A	N/A	-	-	-	-	-	-	5×10^5	-	43
MoS ₂ /HfO ₂ / LiNbO ₃	-1.5 V 500 μ s 22 μ W 7 s	N/A	N/A	-	-	-	-	-	-	40	3×3	44
α -In ₂ Se ₃	\pm 4 V, 0.2 s	N/A	N/A	-	-	-	-	-	-	1000	-	45
TiO ₂	\pm 4 V, 42 ns	0.147 fJ	N/A	-	-	-	-	-	-	10^4	-	46
MoS ₂ /hBN/Gr	\pm 40 V, 1 ms	N/A	N/A	-	-	-	-	-	-	300	-	47
BP/Al ₂ O ₃ /HfO ₂ / Al ₂ O ₃	-4 V-12 V 20 ms	N/A	N/A	\pm 18 V, 20 ms	N/A	N/A	-	-	-	200	3×4	48

PZT/SRO/STO	±3 V, 0.15 ms	N/A	2.6 μs	-1.94 V 10 μs +1.89 V 10 μs	3.1 nJ	1 ns	-	-	-	10 ⁶	3 × 3	49
MoS ₂	±10 V, 10 s	N/A	N/A	±10 V, 10 s	N/A	N/A	-	-	-	-	3 × 3	50
WSe ₂ /Al ₂ O ₃ /HfO ₂	-6 V, 5 ms +4 V, 5 ms	N/A	N/A	±6 V, 5 μs	N/A	5 μs	-	-	-	-	3 × 3	51
CeO ₂ /STO	±10 V, 0.5 s	N/A	N/A	±10 V, 0.5 s	N/A	12 ns	-	-	-	-	3 × 3	52
PZT/SRO	±3.5 V, 0.5 μs	1.77 nJ	10 μs	±10 V, 100 ns	56 fJ	30 μs	-	-	-	10 ⁹	2 × 2	53
Gr/3R-WS ₂ /Gr	±2 V, 0.5 ms	N/A	N/A	±0.05 V, 1 s	N/A	19.2 μs	-	-	-	2000	3 × 3 × 3	54
SWNT@GDY	±0.5 V, 1 ms	50 aJ	N/A	±0.5 V, 1 ms	50 aJ	5 ms	-	-	-	10 ⁵	3 × 3 × 3	55
Gr/CIPS/Gr	±1.5 V, 0.1 s	N/A	N/A	±0.5 V, 0.1 s	N/A	N/A	-	-	-	10 ⁵	10 × 10	56
MoTe ₂ /P(VDF-TrFE)	±39 V, 50 μs	N/A	N/A	±25 V, 10 μs	N/A	1.9 μs	-	-	-	10 ⁶	3 × 3 × 3	57
PEDOT:PSS [®] /perovskite/ C60/bathocuproine	±4 V, 10 s	N/A	N/A	±4 V, 10 s	3.63 fJ	2.75 ns	-	-	-	-	64 × 64	58
TiO _x /ZnO	±3.5 V, 100 ns	2 nJ	N/A	±3.5 V, 100 ns	2 nJ	N/A	-	-	-	10 ⁹	90 × 8	59
MoS ₂ /BN/Se	±30 V, 10 ms	N/A	N/A	±30 V, 10 ms	N/A	N/A	-	-	-	3000	3 × 3	60

VO ₂	3 V, 1 μs	3.9 nJ	30 ns	-	-	-	-	-	-	10 ¹²	15 × 15	61
HfZrO	+3.8 V, 100 μs -2 V, 100 μs	N/A	N/A	-	-	-	-	-	-	2 × 10 ⁷	-	62
Hf _{0.5} Zr _{0.5} O ₂ /SiO ₂	+5 V, 500 ps -3.5 V, 500 ps	0.12 fJ	N/A	-	-	-	-	-	-	10 ⁷	-	63
Si/HfSe ₂	±1 V, 1 ns	N/A	60 ns	-	-	-	-	-	-	2.65 × 10 ⁴	32 × 32	64
TiO _x	4.5 V, 10 μs	N/A	100 μs	-	-	-	-	-	-	5 × 10 ⁶	20 × 20	65
BP/Al ₂ O ₃ /WSe ₂ /h-BN	±14 V	N/A	N/A	-6 V +5 V Persistent voltage pulse	N/A	100 μs 5 ms	-	-	-	55	3 × 3	66
VO ₂	-	-	-	-	-	-	ReLU	65 ns	199.5 pJ	5000	-	25
μ-Si multimodal transistor	-	-	-	-	-	-	ReLU	N/A	N/A	-	-	13
TaO _x memristor circuit	-	-	-	-	-	-	ReLU, Sigmoid, Tanh	N/A	N/A	-	-	67
p(V3D3-co-VI)/Al ₂ O ₃ neural circuit	-	-	-	-	-	-	ReLU	N/A	0.5 pJ	-	-	68
NbO ₂ neural circuit	-	-	-	-	-	-	LIF	500 μs	1.65 pJ	5000	-	69

NbO ₂ neural circuit	-	-	-	-	-	-	LIF	300 μ s	N/A	-	-	70
In ₂ O ₃ BHJ-NbO _x	-	-	-	-	-	-	LIF	110 ns	1.8 nJ	100	5 \times 5	71
MoS ₂ neural circuit	-	-	-	-	-	-	IF	500 ms	2.8 nJ	N/A	3 \times 3	72
Nb/NbTi _x O _y neural circuit	-	-	-	-	-	-	IF	200 μ s	2.4 mW	3 \times 10 ⁸	3 \times 2	73
WSe ₂ /HfO ₂ /Gr/ GDYO/Gr	\pm 1.2 V 20 ns	4.8 fJ	8 ns	\pm 1.2 V 20 ns	4.8 fJ	440 ns	ReLU Sigmoid	8 ns	16 fJ for ReLU 80 aJ for Sigmoid	10 ⁶	5 \times 5 \times 4	This work

V : Voltage pulses for programming. E : Energy consumption for programming. N/A: Not available. “-”: No such feature.

^{a)}PEDOT:PSS refers to poly(3,4-ethylenedioxythiophene):poly(styrene sulfonate).

2) The multi-functional integration heavily relies on heterogeneous integration. This work did not solve this problem, and it is also based on multi-layer heterogeneous integration.

Reply: We appreciate your insightful comment and understand the concern. You are right that our original phrasing “heterogeneous integration” could be misleading. What we intended to express is that **most previous neuromorphic systems require assembling multiple separate devices**, each designed for a single function (e.g., sensing, computing, or activation), which inevitably increases circuit complexity and limits scalability.

In contrast, our MM-SFGM **integrates all three key neuromorphic functions—ISC, IMC, and reconfigurable nonlinear activation—within a single device platform**, eliminating the need for device-level functional assembly. To avoid confusion, we have revised the manuscript by **removing the term “heterogeneous integration”** and using descriptive wording that more accurately distinguishes between previous multi-device systems and our single-device multifunctional approach as following:

Abstract (Page 1, Lines 30–31):

Original: “... rely heavily on heterogeneous integration, increasing system complexity and limiting scalability.”

Revised: “... rely on assembling multiple discrete devices to realize different operations, which increases system complexity and limits scalability.”

Introduction (Page 3, Lines 75–76):

Original: “... breaks from the traditional need for heterogeneous integration, offering a compact, reconfigurable, and general-purpose solution ...”

Revised: “... eliminates the traditional need to combine multiple device types for different functions, offering a compact, reconfigurable, and general-purpose solution ...”

Discussion/Conclusion (Page 19, Lines 430–431):

Original: “... while circumventing the limitations of conventional heterogeneous silicon integration.”

Revised: “... while circumventing the limitations of conventional multi-device silicon architectures.”

3. The manuscript does not provide a clear description of the quantitative breakthroughs in key performance indicators.

Reply: We sincerely thank you for this insightful comment. We fully agree that our

work does not aim to demonstrate quantitative superiority in performance metrics such as speed, energy consumption, or endurance compared to state-of-the-art commercial or research processors. Instead, the core contribution of this work is the first **qualitative demonstration** of a single device that monolithically integrates three essential neuromorphic operations—in-sensor computing (ISC), in-memory computing (IMC), and reconfigurable nonlinear activation functions (NAFs)—within a unified platform.

While Supplementary Table 4 provides quantitative comparisons, showing competitive energy efficiency (fJ-level) and nanosecond-scale operation, we emphasize that these are supporting characteristics, not the primary claims. The main focus of this study is the conceptual and architectural breakthrough of merging sensing, memory, and activation in a single compact system, which reduces circuit complexity and lays the foundation for future quantitative optimization via device- and circuit-level engineering.

To make this explicit, we have revised the Discussion section (Pages 18–19, Lines 402, 407–413, 425–427) to clearly state that this work represents a **qualitative milestone** in neuromorphic functional integration, and that achieving quantitative superiority will require further engineering development in future studies.

Reviewer 1:

Editor, I'm good with the paper, and I think peer review has done its job. I recommend you send it to print, and I don't need to see it again.

I'm impressed by Fig. 3e. I suggest you make it larger. Optionally, I also suggest you add a couple of additional data sets to it, especially from Fig. 3c of this paper: <https://arxiv.org/abs/2505.15936>. You will notice that your data offers lower conductances for a given error/standard deviation compared to the paper I referenced (meaning that you offer lower power consumption for a given error), as well as extending the range of the data from the referenced paper. In addition, your data offers both lower conductances as well as lower errors compared to most other published literature.

Reply: We sincerely thank you for the highly positive evaluation of our manuscript and for the strong recommendation for publication. We greatly appreciate your time and encouragement.

Following your suggestion, we have revised **Fig. 3c** by incorporating the additional datasets from the referenced work (<https://arxiv.org/abs/2505.15936>). This comparison further highlights the advantages of our results in terms of conductance and error characteristics. We are grateful for your insightful comments and constructive suggestions, which have helped us improve the clarity and impact of the manuscript.

Reviewer 2:

I think the authors has provided a large amount of experimental data, which proves that a great deal of time and effort were invested in this work. However, the biggest obstacle hindering the development of this article lies in the appropriate organization. And I still believe that this work needs to be major revised and the article organization needs to be re-done before it can be published.

My previous objection to this work's publication on Nature Communications was related to two insufficient points: (1) the novelty of the research; (2) whether the key scientific issues identified by the author as needing to be addressed in the research have been successfully resolved through innovation.

In the author's response, it has been clearly stated that the focus of the research lies in the fact that these three functions (in-sensor computing, in-memory computing and multiple nonlinear activation functions) can be achieved on a single device, rather than in the pursuit of a breakthrough in a specific single performance. The key scientific question that the author aims to address is existing works not compact, energy-efficient enough. So, what was the final result? How compact and energy-efficient? The images captured by the mm-scale optical microscope are not compact, and there is no discussion regarding energy efficiency. I totally agree with Reviewer 1 that not all tasks require quantification, but even for the qualitative study on "How compact and energy-efficient", there is no such discussion in this article on these core issues. Among the 5 figures in the manuscript, 1 figure is about the innovation of the device structure, 3 figures are showing the implementation of a single function, and the last figure only show the demonstration of function integration but no comparison. None about "How compact and energy-efficient". This led to my previous question, which is that the innovation of the device structure and single performance have not been adequately demonstrated to support the publication of the paper. And it also confused me that although traditional split-gate flash (reported by a large amount of works) can also achieve these integration functions, they simply haven't conducted these tests.

In conclusion, I believe it is necessary to clearly define the core issues that the research needs to address and provide sufficient data support or discussion on the core issues.

Supplementary Tables 3 and 4 present comparisons of energy consumption and area efficiency. Although this work does not achieve the best performance among the compared studies, it still remains at a leading level. Additionally, my original suggestion was for the authors to highlight these comparisons in the main text, as they represent the innovative aspects emphasized in the title of their work. Therefore, while

the data provided in the Supplementary Information (SI) can meet the requirements of the question, it would be better to include these comparisons in the main text to better reflect these aspects.

Reply: We sincerely thank you for the careful reading of our manuscript and for the thoughtful and detailed comments. We appreciate your emphasis on clearly defining the core scientific questions and ensuring that the manuscript directly addresses them with sufficient data and discussion.

We fully agree that the central motivation of this work is not to pursue record-breaking performance in a single metric, but to address a key system-level challenge in neuromorphic hardware: how to achieve compact and energy-efficient integration of sensing, linear computing, and nonlinear activation. In the revised manuscript, we have substantially reorganized the Discussion section to make this core issue explicit and to directly answer the questions of “how compact” and “how energy-efficient” the proposed platform is.

Specifically, we have now explicitly introduced and discussed Table 2 in the main text, which benchmarks key parameters of nonlinear activation devices and circuits, including area and energy consumption. This table shows that conventional activation implementations—often based on digital processors or analog CMOS circuits—occupy substantial area and consume significant power, frequently dominating the system footprint. In contrast, the proposed MM-SFGM implements ReLU- and Sigmoid-type activation natively within the same device used for sensing and memory, resulting in femtojoule- to attojoule-level activation energy and a compact device footprint. We further clarify that the reported device area is obtained using laboratory-scale fabrication, and that advanced nanolithography would allow the footprint to be significantly reduced, providing substantial room for future scaling.

In addition, we have revised the Discussion to emphasize that the compactness discussed in this work refers to functional density at the device and array level, rather than the size of the experimental optical or electronic setup. By consolidating sensing, linear computation, and nonlinear activation into a single device, the MM-SFGM eliminates external activation circuits and repeated data movement between heterogeneous modules, thereby reducing system-level energy consumption and latency. These system-level benefits are now explicitly discussed and supported by Table 1, which compare energy consumption and area efficiency with representative platforms.

Following your suggestion, we have moved Supplementary Tables 3 and Table 4 from the Supplementary Information into the main text (Tables 1 and 2) and added corresponding discussion to highlight the relevant comparisons.

We sincerely appreciate your constructive feedback, which has significantly improved the clarity, organization, and focus of the manuscript. We believe the revised version now more clearly articulates the core scientific question and provides sufficient data and discussion to support the conclusions.